# A whole-organism landscape of X-inactivation in humans

Bjorn Gylemo, Maike Bensberg, Colm E Nestor*

Crown Princess Victoria Children´s Hospital, Department of Biomedical and Clinical Sciences (BKV), Linköping University, Linköping, Sweden

## eLife Assessment

The study provides a **valuable** analysis of escape from X-inactivation based on three rare female GTEX-donors with non-mosaic X-inactivation. The methods and analyses are **solid** and broadly support the authors' claims. Their data are more comprehensive than those presented previously and add significant weight to evidence for which genes are inactivated or escape from X inactivation in humans.

## Abstract

As females are mosaic for X-inactivation, direct determination of X-linked allelic expression in bulk tissues is typically unfeasible. Using females that are non-mosaic (completely skewed) for X-inactivation (nmXCI) has proven a powerful and natural genetic system for profiling X-inactivation in humans. By combining allele-resolution data for one previously reported and two newly identified nmXCI females, we directly determined X-inactivation status of 380 X-linked genes across 30 normal tissues, including 198 genes for which XCI status is directly determined for the first time. Our findings represent a substantial advance in our understanding of human X-inactivation and will serve as a reference for dissecting the genetic origin of sex bias in human traits. In addition, our study reveals nmXCI as a common feature of the human female population, with profound consequences for the penetrance and expressivity of X-linked traits in humans.

*For correspondence:
colm.nestor@liu.se

Competing interest: The authors declare that no competing interests exist.

## Introduction

During embryonic development in humans, female cells (46, XX) inactivate a single X-chromosome to balance the dosage of X-linked gene expression between females (XX) and males (XY). The process of X-chromosome inactivation (XCI) begins during the peri-implantation stage in humans and is initiated by expression of the long non-coding RNA (lncRNA) *XIST* from the X inactivation center (XIC) of one X-chromosome in each female cell. *XIST RNA* coats the X-chromosome which in turn recruits protein complexes, including chromatin modifiers which establish a facultative heterochromatin state. Silencing histone modifications and the deposition of DNA methylation ultimately result in the transcriptionally silent inactive X-chromosome (Xi) (*Figure 1—figure supplement 1A*). The selective expression of *XIST* from only one X-chromosome needs to be tightly regulated. Whereas the exact mechanism by which a given X-chromosome in a female cell is selected for inactivation is unknown, the XIC in humans includes several lncRNAs that have been implicated in the regulation of *XIST* expression and X-inactivation (*Figure 1—figure supplement 1A*; *Rosspopoff et al., 2023*). Although, the initial choice of X-chromosome for inactivation is random, the inactivated X-chromosome is stably inherited in a clonal fashion throughout all subsequent cell divisions (*Patrat et al., 2020*). Interestingly, the process of X-inactivation (XCI) is incomplete; approximately 15% of genes escape XCI and remain expressed from both X-chromosomes (*Balaton et al., 2015*). These 'escape' genes are consequently more highly expressed in females than in males, although the degree of sex-biased expression of

**Figure 1.** Identification of females with non-mosaic X-inactivation. (**A**) The patterns of X-chromosome inactivation (XCI) in women resulting in mosaic (right female) or non-mosaic XCI (nmXCI). The presence of genetic variants can result in nmXCI females by (**i**) directly determining which X-chromosome can be inactivated (primary skewing, left female) or by (**ii**) imparting a selective advantage to a small number of cells (secondary skewing, middle female). Xa, active chr X; Xi, inactive chr X, Xm, maternal chr X, Xp, paternal chr X. (**B**) Single-tissue median allelic expression (AE) and standard error of all nonPAR genes on chromosome X (chr X) not previously classified as variable in all 285 women in Genotype-Tissue Expression (GTEx). (**C**) Allelic expression per tissue of non-PAR chr X genes not previously classified as variable in mosaic females (median allelic expression <0.475) and three females identified as non-mosaic, nmXCI-1, nmXCI-2, and UPIC (median allelic expression >0.475). Boxplot indicating median, 25th and 75th percentiles. (**D**) Copy number as log2 ratio of chromosome 17 (chr 17) for nmXCI females, UPIC, nmXCI-1, and nmXCI-2. Trisomy 17 p in UPIC is highlighted.

The online version of this article includes the following figure supplement(s) for figure 1:

**Figure supplement 1.** Characterisation of Trisomy 17p in one non-mosaic female.

escape genes can vary and may be greater for those genes that lack a functional homologue on the Y-chromosome (*Carrel and Willard, 2005*). The sex-biased expression of escape genes has been suggested to mediate sex differences in both the frequency and severity of several diseases (*Souyris et al., 2018*; *Mousavi et al., 2020*). In addition, genes in the pseudoautosomal regions (PAR1 and PAR2) of the X-chromosome also remain expressed from the Xi and are thus always bi-allelically expressed. PAR1 and PAR2 are short regions of homology between the X- and Y-chromosomes that are essential for pairing of the sex chromosomes during meiosis in males (*Helena Mangs and Morris, 2007*). In contrast to escape genes, which tend to be more highly expressed in female cells, genes in the PARs have been suggested to show male-biased expression (*Tukiainen et al., 2017*). Consequently, the terms PAR and non-PAR are used to distinguish between the biallelic expression (escape) of genes within the PARs and those located elsewhere on the X-chromosome.

Human females are typically 'mosaic' for X-inactivation; female tissues contain a mixture of cells that have inactivated either the maternal or paternal X-chromosome (*Figure 1A*). Mosaicism renders analysis of XCI in primary human tissues highly complex or unfeasible. Consequently, our knowledge of XCI and escape across human tissues is surprisingly sparse and typically based on indirect measures of XCI, such as DNA methylation, sex-biased expression, or from observations in cell lines and animal models (*Carrel and Willard, 2005*; *Cotton et al., 2013*). In a seminal study, Brown and colleagues reported XCI status for 639 genes by integrating data from three published studies that each used different indirect approaches and model systems to infer XCI status (*Balaton et al., 2015*). This resource continues to serve as a valuable reference for inferred XCI status in humans.

Interestingly, rare cases of humans in which the same parental X-chromosome has been inactivated in all cells (non-mosaic XCI, nmXCI) have been reported (*Tukiainen et al., 2017*; *Plenge et al., 1997*; *Amos-Landgraf et al., 2006*; *Shvetsova et al., 2019*). Organism-wide nmXCI can arise from a constitutional genetic variant that directly interferes with the inactivation of a specific parental X-chromosome, such as mutations in the promoter region of the *XIST* gene (*Plenge et al., 1997*) and X-autosome translocations (*Favilla et al., 2021*; *Hatchwell et al., 1996*). Disruption of the XIC locus or autosomal regions involved in the choice of the X-chromosome to be inactivated has similarly been suggested to result in direct nmXCI, so called 'primary' skewing (*Clerc and Avner, 2006*). Alternatively, selection against any deleterious constitutive genetic variant would result in clonal expansion of cells and indirectly result in nmXCI; 'secondary' skewing (*Figure 1A*). As nmXCI females lack the confounding effect of mosaicism, they represent a powerful system to study human XCI by enabling direct determination of XCI status from bulk tissue samples. Indeed, using a single nmXCI female identified in the Genotype-Tissue Expression (GTEx) database (*Lonsdale et al., 2013*), Tukiainen and colleagues generated the first comprehensive reference map of XCI status across human tissues based on direct determination of allele-specific expression of X-linked genes (*Tukiainen et al., 2017*). However, as the ability to determine allele-specific expression is limited to the presence of an informative (heterozygous) genetic variant in the gene of interest, the XCI status of just 186 genes was determined across tissues in this one nmXCI female.

In this study, we identify two additional unrelated nmXCI females in the GTEx database. By combining allele-resolution data for the one previously reported nmXCI female and two newly identified nmXCI females, we directly determined X-inactivation status of 380 X-linked genes across 30 normal tissues, including 15 tissues in which XCI has not previously been characterized. This unique dataset allowed investigation of tissue-specific escape from XCI in humans and generation of the most extensive multi-tissue map of human X-inactivation to date.

## Results

### Non-mosaic X-inactivation is common in human females

Whereas the frequency of nmXCI females in the general population was originally thought to be less than 1:500 (*Amos-Landgraf et al., 2006*), more recent work has suggested a frequency of as high as 1:50 (*Shvetsova et al., 2019*). To identify nmXCI females, we screened a single tissue of all 285 female donors*Supplementary file 1A-B* in the GTEx database (v8 release) (*Lonsdale et al., 2013*). Calculating the allelic expression (AE, how much the ratio of Xa/Xi reads deviates from the expected 0.5) of all X-linked genes outside of the PARs (i.e. the nonPAR AE) allows assessment of skewing in female tissues. Briefly, allelic expression was determined by calling heterozygous SNPs on the X-chromosome using genomic data (whole-exome sequencing (WES) or whole-genome sequencing (WGS)) and subsequently counting the number of each base at every heterozygous SNP (hetSNP) position in RNA-seq data. The allelic expression can then be calculated as abs(0.5 - (reference reads/ total reads)), where a value of zero indicates complete biallelic expression and a value of 0.5 indicates complete monoallelic expression. We specifically exclude PAR genes from this calculation as they are not subject to XCI, and consequently, their Xa/Xi read count ratio is not significantly affected by XCI skewing. Three female samples showed an exceedingly high degree of skewing (median chr X nonPAR allelic expression >0.475 i.e., less than 2.5% of reads coming from the 'inactive' allele), consistent with expression from a single parental chromosome, nmXCI (*Figure 1B*). Extending allelic expression analysis to all available tissues across the three candidate females*Supplementary file 1C* confirmed their complete nmXCI status (*Figure 1C*). These three nmXCI individuals included the single previously identified nmXCI female, UPIC (*Tukiainen et al., 2017*), confirming the accuracy of our screening approach (*Figure 1B–C*).

nmXCI may result from deleterious genetic mutations causing primary (*Plenge et al., 1997*) or secondary (acquired) skewing (*Clerc and Avner, 2006*) or from stochastic non-random XCI (*Shvetsova et al., 2019*). Whereas no small-scale mutations in the XIC were identified in any of the nmXCI females, trisomy 17 p was observed in UPIC (*Figure 1D*, *Figure 1—figure supplement 1B*). How trisomy 17 p could result in non-random X-inactivation is unclear, but an unbalanced 17 p:X translocation would result in strong selection for silencing of the abnormal X and consequential nmXCI (*King et al., 1998*). However, the expression of genes on 17 p was consistently higher than those on 17q across

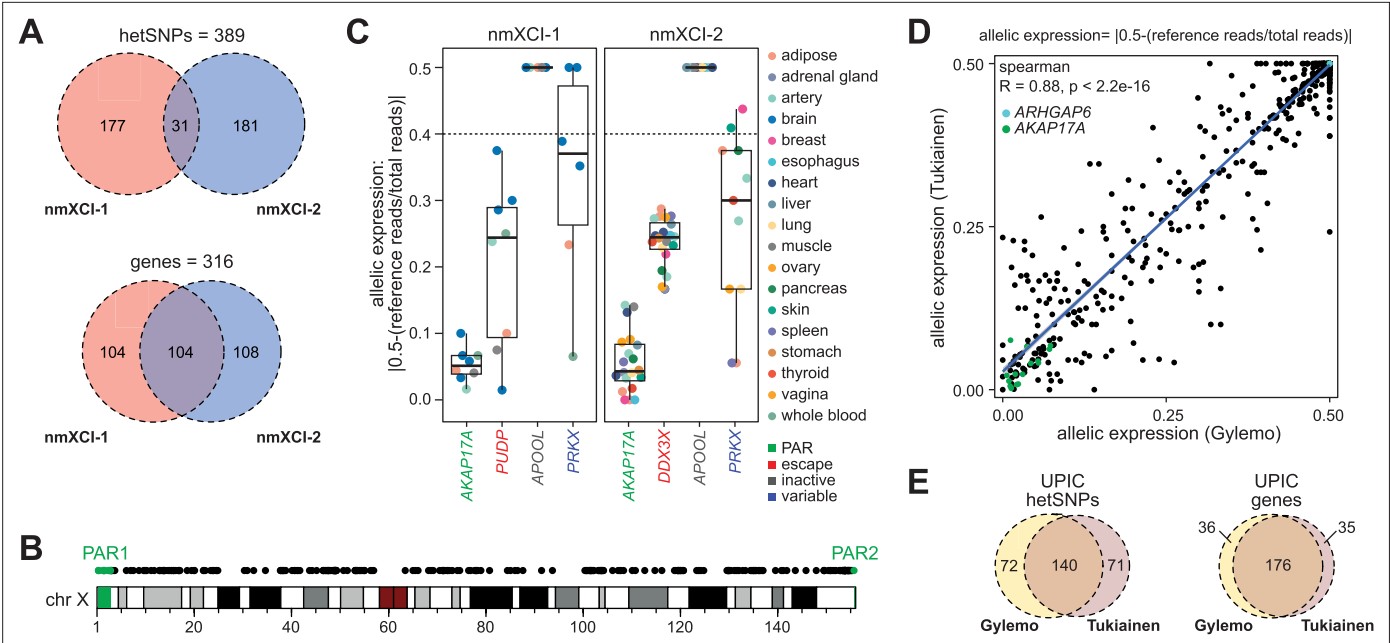

**Figure 2.** Characterization of two novel human non-mosaic (completely skewed) for X-inactivation (nmXCI) females. (**A**) Overlap of genic heterozygous SNPs (hetSNPs) (upper) and genes with hetSNP (lower) across the two novel nmXCI females (nmXCI-1 and nmXCI-2). Genic hetSNPs were identified using both whole-exome sequencing (WES) and whole-genome sequencing (WGS) for each individual. (**B**) Distribution of assessed genes across the X-chromosome. Genes located in the pseudoautosomal region 1 (PAR1) and PAR2 are highlighted in green. (**C**) Allelic expression per tissue for well-characterised PAR, (*AKAP17A*), escape (*PUDP, DDX3X*), inactive (*APOOL*), and variable (*PRKX*) genes (*Tukiainen et al., 2017*). Boxplot indicating median, 25th and 75th percentiles. (**D**) Spearman correlation of allelic expression values using our analysis approach (Gylemo) and the Tukiainen et al analysis pipelines for female UPIC. (**E**) Overlap of genic hetSNPs (left) and genes (right) identified by our analysis (Gylemo) and the Tukiainen et al analysis pipeline in female UPIC.

The online version of this article includes the following figure supplement(s) for figure 2:

**Figure supplement 1.** Comparison of allelic expression values using different analysis approaches.

all tissues in UPIC, suggesting the duplicated 17 p was not silenced (*Figure 1—figure supplement 1C*). Indeed, further investigation of donor metadata in GTEx revealed that UPIC had been diagnosed as mosaic for trisomy 17 p by traditional karyotyping but did not provide a detailed karyotype. Interestingly, whereas all three females were nmXCI, females nmXCI-1 (GTEx ID: 13PLJ) and nmXCI-2 (GTEx ID: ZZPU) showed greater variation in allelic expression across tissues than UPIC (*Figure 1C*), suggesting that their nmXCI may have resulted from clonal selection driven by constitutive genetic variants (secondary skewing).

## A landscape of X-inactivation across normal human tissues

The ability to detect allele-specific expression is dependent on the presence of a heterozygous genetic variant (SNP) in the gene of interest. Using WES and WGS for both newly discovered nmXCI donors, we identified 389 heterozygous SNPs across 316 X-linked genes (count and overlap between nmXCI-1 and nmXCI-2 of heterozygous SNPs and genes in *Figure 2A*, distribution of genes covered along the X-chromosome in *Figure 2B*, *Supplementary file 1D*). Allelic expression ratios were highly consistent between nmXCI females and allowed us to identify known escape (PAR and nonPAR), inactive, and variable (*Tukiainen et al., 2017*) XCI genes (*Figure 2C*). Next, we applied our analytical workflow to the previously identified nmXCI sample, UPIC, observing that the allelic expression levels seen using our analysis pipeline largely matched the findings of (*Figure 2D*, *Figure 2—figure supplement 1A*; *Tukiainen et al., 2017*). Notably, our inclusion of additional WGS data of UPIC and identification of 76 further SNPs allowed determination of allelic expression of 36 further genes in this individual (*Figure 2E*). The use of updated gene models, correction of reference alignment bias, and more stringent quality filtering resulted in the exclusion of 35 genes for which allelic expression was erroneously reported (*Figure 2E*, *Figure 2—figure supplement 1B*).

Next, we integrated data from all three nmXCI individuals (nmXCI-1, nmXCI-2, and UPIC), allowing direct allelic expression detection of 380 X-linked genes across 30 tissues, including 15 tissues for which XCI status has not been directly assessed previously. While 195 genes and 17 tissues could only be examined in a single individual, 185 genes and 13 tissues were examined across multiple nmXCI females, allowing for selected inter-tissue and inter-individual comparisons (*Figure 3A–B*). Classification of allelic imbalance (mono-allelic|bi-allelic) of individual alleles was initially determined by the binomial test ($P_{ADJ}$ <0.01), according to best practice (*Castel et al., 2015*). However, upon visual inspection, we identified several well-characterised escape and inactive genes that may have been misclassified as variable escape (*Figure 3C*). Inappropriate classification of allelic imbalance results from the well-documented sensitivity of the binomial test to data over-dispersion, which frequently occurs with read count data (*Castel et al., 2015*; *Chen et al., 2016*). Consequently, we further manually curated the allelic expression status of all 380 genes using the empirical guideline of an allelic ratio >0.4 as indicating mono-allelic expression and the potential consequences of high and low read counts. Our three criteria for manual re-classification were (i) *low power*, indicating genes with a low statistical power but consistent escape pattern, (ii) *low read count*, indicating genes with a consistent escape pattern but with non-significant escape in a single tissue, and (iii) *over-estimation*, genes in which a high read counts inflated the binomial p-value. (*Figure 3C*, *Figure 3—figure supplement 1*, *Supplementary file 1D-E*). This manual curation resulted in the re-classification of allelic expression status of 32 genes (*Supplementary file 1E*, *Figure 3—figure supplement 1*). Our classifications of XCI status based on direct determination of allele-specific expression represent one of the most extensive and high-confidence maps of X-inactivation to date (*Figure 3D–E*, *Figure 3—figure supplement 2*). Allelic expression for all genes independent of XCI status was highly correlated between individuals (*Figure 4—figure supplement 1A*), and by including three nmXCI individuals in our analysis, we were able to determine cross-tissue XCI status for 13 genes which were only covered in one tissue in the previous assessment based on UPIC alone (*Figure 4A*). Interestingly, whereas we reveal *EGFL6*, *TSPAN6,* and *CXorf38* as variable escape genes, these were previously classified (Balaton et al) as inactive, variable, and escape, respectively. Finally, we compared our classification with the classification of X-linked escape status from donor UPIC as determined by Tukiainen and colleagues (*Carrel and Willard, 2005*; *Cotton et al., 2013*; *Shvetsova et al., 2019*). As expected, our classification was largely consistent with that of Tukiainen et al., but also revealed the possible misclassification of XCI status of several genes and further added the XCI status for 198 genes for which XCI status had not previously been directly assessed (*Figure 4B*). Our direct assessment of allelic expression further allowed for validation of XCI status previously determined by indirect measures (*Balaton et al., 2015*; *Tukiainen et al., 2017*). We confirmed XCI status for many previously classified inactive and escape genes, whereas most reported variable escape genes are reclassified as inactive based on our analysis (*Figure 4—figure supplement 1B–C*). However, as the XCI classifications reported here are largely based on inter-tissue comparisons within individuals, most previous classifications were based on inter-individual comparisons of XCI, confounding direct comparisons between studies.

## Discussion

Here, we describe an extensive multi-tissue map of human X-inactivation directly determined from allele-specific expression of X-linked genes. This data represents a doubling of the number of X-linked genes for which XCI status has been directly determined across multiple normal tissues and individuals. Our results suggest that XCI is stable both within and across individuals and that tissue-specific escape from X-inactivation is rare (11.6% of total genes (44 variable genes/380 total covered X-linked genes)) and often challenging to characterize. Variability in X-linked gene expression is important to characterize, as XCI variability can directly modify the expressivity of X-linked pathogenic variants. This provides a mechanism for sex-biased expression differences in certain biological contexts and tissues, which in turn can modify disease risk.

nmXCI females allow for direct determination of allele-specific expression of X-linked genes, even in bulk tissue samples. Our study further highlights the power of such natural genetic systems to shed light on XCI in humans. Whereas the origin of nmXCI in females is typically unknown, nmXCI females likely harbor one or more mutations that (i) directly affect the choice of X-chromosome to be inactivated (primary skewing) or (ii) result in selection against cells carrying an active mutated X-chromosome (secondary/acquired skewing). As such, the X-linked genetics of nmXCI females is clearly

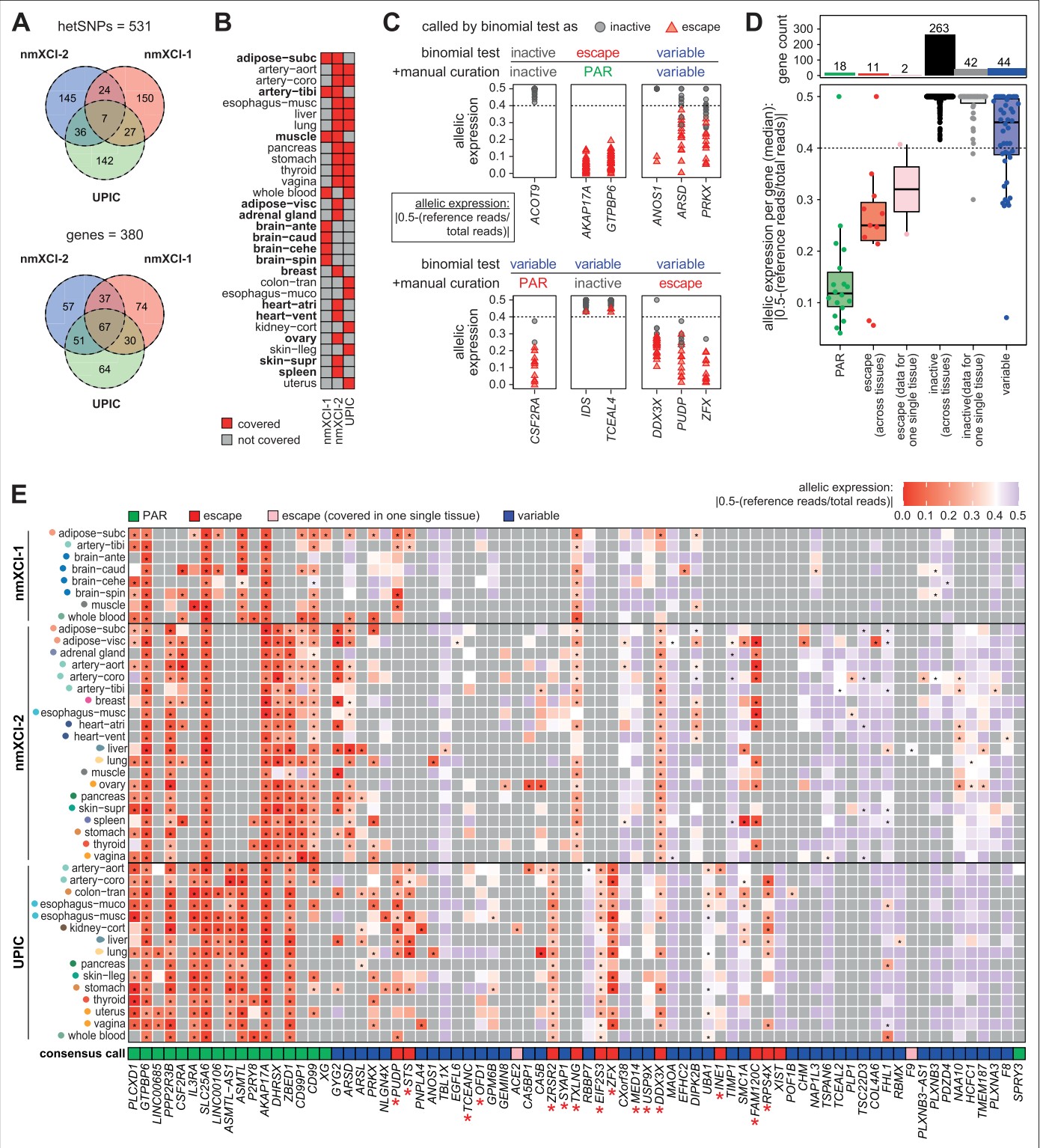

**Figure 3.** An extended landscape of X-inactivation in humans. (**A**) Overlap of genic heterozygous SNPs (hetSNPs) (upper) and genes with hetSNP (lower) across the three females (nmXCI-1, nmXCI-2, and UPIC) with non-mosaic X-inactivation (nmXCI). (**B**) Tissues covered in each nmXCI female. Tissues not covered in UPIC are indicated in bold. (**C**) Examples of genes classified by the binomial test alone and after manual curation. Allelic expression across tissues is shown with X-chromosome inactivation (XCI) status based on the binomial test indicated as inactive (grey circle) or escape (red triangle). (**D**) Allelic expression of X-linked gene categories (lower) and number of genes included in each category (upper). Genes classified as escape and inactive are separated based on whether allelic expression was determined across multiple tissues or in a single sample (data for one single tissue). Boxplot

*Figure 3 continued on next page*

*Figure 3 continued*

indicating median, 25th and 75th percentiles. (**E**) Heatmap showing the allelic expression of all genes that show constitutive or variable escape from XCI. Black asterisks within the tiles indicate a significant expression from the inactive X-chromosome (i.e. escape, FDR-corrected binomial q-value <0.01). The 'consensus call' tile is the assigned XCI status across tissues and individuals for each gene with genes classified as variable, including both intra- and interindividual variation. Red asterisks indicate genes in which manual curation of XCI status was performed. Grey tiles indicate missing data. (**B, E**) Tissue abbreviations can be found in *Supplementary file 1G*.

The online version of this article includes the following figure supplement(s) for figure 3:

**Figure supplement 1.** Read counts for minor and major alleles for all genes where the X-chromosome inactivation (XCI) status was manually curated.

**Figure supplement 2.** Heatmap showing the allelic expression of all X-linked genes assayed in this study.

atypical and could result in aberrant establishment and maintenance of XCI in these females. Whereas the highly similar patterns of XCI observed across the three genetically independent nmXCI females analyzed here suggest that such effects are subtle, additional multi-tissue studies of XCI in mosaic females are required to confirm the wider applicability of our findings. Finally, our identification of two additional nmXCI females among the 285 females in the GTEx database further substantiates recent claims of extreme skewing of XCI as a frequent and major modifier of trait penetrance and expressivity in the human population (*Shvetsova et al., 2019*; *Roberts et al., 2022*).

## Materials and methods
### Allele-specific expression analysis for initial identification of nmXCI females

For the initial screen to identify nmXCI females in GTEx, pre-processed SNPs were called from WES data from GTEx (GTEx_Analysis_2017-06-05_v8_WholeExomeSeq_979Indiv_VEP_annot.vcf.gz, see https://gtexportal.org/home/methods for specifics on data processing). Downloaded data was lifted over from hg19 to hg38 with LiftoverVcf (GATK v.4.1.9.0), and bcftools (v1.10.2) (*Danecek et al., 2021*) was utilized to include only hetSNPs on the X-chromosome. One RNA-seq sample per female was downloaded from AnVIL using the Gen3 client, as aligned BAM files (see https://gtexportal.org/home/methods for specifics on data processing). Tissues used for the screen were selected based on abundance in the entire data set (muscle, n=239; skin-leg, n=21; thyroid, n=1; adipose-subc, n=2; artery-tibi, n=2; esophagus-muco, n=2; nerve, n=2; adipose-visc, n=1; ovary, n=1; uterus, n=1; see

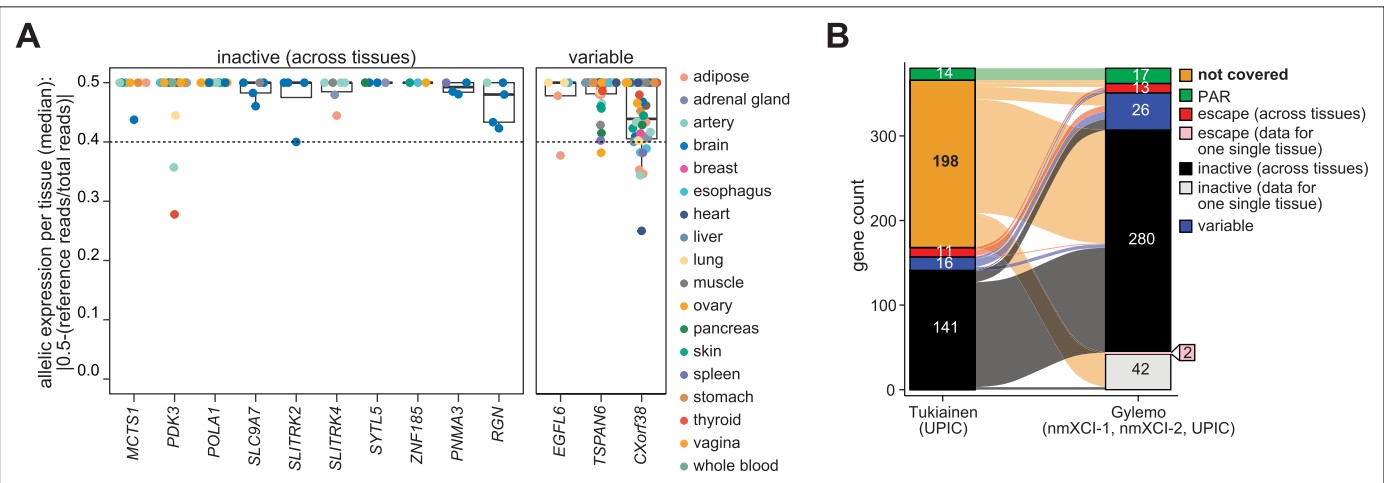

**Figure 4.** Classification and novel X-chromosome inactivation (XCI) assessment of X-linked genes. (**A**) Allelic expression across all available tissues in all three non-mosaic X-inactivation (nmXCI) females for genes which were only covered in one tissue in the previous assessment based on UPIC alone (*Tukiainen et al., 2017*). (**B**) Alluvial plot showing classification of escape status of X-linked genes based on our analysis (Gylemo) compared to a previous assessment based on UPIC alone (Tukiainen) (*Tukiainen et al., 2017*). Genes classified as escape and inactive are separated based on whether allelic expression was determined across multiple tissues or in only one sample (data for one single tissue).

The online version of this article includes the following figure supplement(s) for figure 4:

**Figure supplement 1.** Re-classification of XCI status of X-linked genes.

*Supplementary file 1H* full list of abbreviations). Following data download, allelic expression (AE) analysis of allele counts for hetSNPs was retrieved from RNA-seq data using GATK ASEReadCounter (v.4.1.9.0), requiring a minimum base quality of 20 in the RNA-seq data. To remove potentially spurious sites, conservative filters were applied to the data: variant call read depth in WES≥20 per allele, minor allele read count >10% of the total read count, and RNA-seq read depth >10 reads. If a gene carried multiple hetSNPs, only the one with the highest RNA-seq read count was included. AE data from the screen can be found in *Supplementary file 1B-C*. XCI skew of each individual participant was assessed by calculating the single-tissue median AE of all non-PAR chr X genes not detected previously as variable (*Tukiainen et al., 2017*; *Supplementary file 1A*) in all 285 women in GTEx. Median chr X nonPAR AE of higher than 0.475 (equates to <2.5% of reads coming from the inactive X-chromosome) indicates expression from a single parental chromosome and was used as a cutoff to distinguish between mosaic and non-mosaic XCI (nmXCI) females. As a control, UPIC was included. UPIC is a female which has previously been shown to have nmXCI (*Tukiainen et al., 2017*). As UPIC is not included in GTEx_Analysis_2017-06-05_v8_WholeExomeSeq_979Indiv_VEP_annot.vcf.gz, SNP calling was performed using WES data and AE analysis was performed, ultimately leveraging her AE data as a positive control.

## Detailed analysis of allele-specific expression and XCI status in nmXCI females

For further assessing XCI status of the three females with nmXCI (high median chr X nonPAR AE), the remaining RNA-seq tissue samples available for all three donors were downloaded from GTEx and processed as described above.

We further downloaded WES and whole-genome sequencing (WGS) data available for the three nmXCI females, UPIC, 13PLJ (nmXCI-1) and ZZPU (nmXCI-2), and performed SNP calling. Briefly, WES BAM and WGS CRAM files for each donor were downloaded from AnVIL using the Gen3 client. The files were sorted and converted into FASTQ files using Samtools (v1.9) (*Danecek et al., 2021*). Raw FASTQ reads were quality trimmed using FastP (v.0.20.0) (*Chen et al., 2018*) with default settings. BWA-MEM (*Li and Durbin, 2010*) was used for mapping reads to the human genome build 38 (hg38, GCA_000001405.15_GRCh38_no_alt_analysis_set.fna) using default settings. Sambamba v0.8.2 (*Tarasov et al., 2015*) was used to mark duplicate reads. The GATK best practice germline short variant discovery pipeline was used to process the aligned reads, using base quality score recalibration and local realignment at known insertions and deletions (indels) (GATK v.4.2.6.1). Indels and SNPs were called jointly across all samples, for both WGS and WES data. Default filters were applied to indel and SNP calls using the variant quality score recalibration (VQSR) approach of GATK. All RNA-seq samples available for each participant (*Supplementary file 1F*) were downloaded from AnVIL as aligned BAM files using the Gen3 client. Samtools (v1.9) (*Danecek et al., 2021*) was used to sort and convert the BAM files to FASTQ files and quality trimming was done with FastP (v.0.20.0) (*Chen et al., 2018*) with default settings. RNA-seq reads were aligned with STAR 2-pass mode (v2.7.10a) (*Dobin et al., 2013*) with GCA_000001405.15_GRCh38_no_alt_analysis_set.fna and `--sjdbGTFfile` gencode.v40.annotation.gtf index. WASP filtering was performed to reduce reference bias (*van de Geijn et al., 2015*). Briefly, GATK SelectVariants was used to extract hetSNPs (that passed the VQSR filtering) for each individual from either the WGS or WES VCFs, and subsequently passed to STAR via `--varVCFfile` and–waspOutputMode. Reads failing WASP filtering were removed. Following data pre-processing, AE analysis of the allele counts for hetSNPs was retrieved from RNA-seq data using GATK ASEReadCounter. Heterozygous variants that passed VQSR filtering were first extracted for each sample from WES and WGS VCFs using GATK SelectVariants. Following this, sample-specific VCFs and RNA-seq BAMs were input to ASEReadCounter requiring a minimum base quality of 20 in RNA-seq data. ASEReadCounter outputs from WGS and WES pipelines were joined. If overlapping sites were identified, the hetSNP read counts were merged, and the AE counts from the assay with the highest total read count was kept. To remove potentially spurious sites, conservative filters were applied to the data (variant call read depth ≥ 10 per allele and minor allele read count >10% of the total variant read count for WES/WGS data, and an RNA-seq read depth of more than seven reads). If a gene carried multiple hetSNPs, the hetSNP detected in the highest number of tissues was kept. If two hetSNPs were detected in the same number of tissues, the one with the highest RNA-seq read count was kept. The 'pituitary' sample from nmXCI-1 was excluded as it had a lower median chr X

nonPAR AE than all other tissues from the same individual. Furthermore, the 'lymphoblasts' sample was excluded from donor UPIC since it is unclear how EBV transformation of lymphocytes affects the X-chromosome. The final ASE table for all X-linked genes in the nmXCI females can be found in *Supplementary file 1D*.

### X-inactivation status categorization

Tissue-specific X-inactivation status categorization was performed as previously described (*Tukiainen et al., 2017*). Briefly, XCI status of genes was assessed by comparing the allelic read count ratios at each filtered X-chromosomal hetSNP, in each tissue individually. Whether there was significant expression from the inactive X-chromosome (escape) at each hetSNP was tested with a one-sided binomial test where the reads from Xi compared to the total read count were expected to be significantly greater than 0.025 or 2.5% (hypothesized probability of success = 0.025). FDR correction was applied to p values from the binomial test for each of the tissues separately using the rstatix R package version 0.7.2Kassambara A. 2023. q values <0.01 were considered significant and indicative of XCI escape for the hetSNP and tissue.

To assess across-tissue XCI status of genes, we leveraged the q values obtained above. If a given hetSNP showed significant Xi expression in all tissues in which the gene is expressed, the SNP was categorized as 'escape (across tissues),' a hetSNP with non-significant Xi expression across all tissues in which the gene is expressed was classified as 'inactive (across tissues).' If a gene was significant in more than one but not in all tissues in which it is expressed the gene was classified as 'variable.' Genes for which we had AE data in only one tissue were either labeled 'inactive (data for one single tissue)' or 'escape (data for one single tissue)' based on the binomial test. Finally, genes residing in the pseudo-autosomal region (PAR) were labeled 'PAR.' Upon inspection of the classification results, several well-characterized escape genes and inactive genes were misclassified as variable escape. Consequently, we manually curated the allelic expression status of all 380 genes using the empirical guideline of an allelic ratio >0.4 as indicating mono-allelic expression while also considering the consequences of high and low read counts. Our three criteria for manual re-classification were low read count, low power, and over-estimation, indicating genes with a low statistical power but consistent escape pattern (low power), consistent escape pattern but with non-significant escape in single tissue (low read count) and over-estimation due to high read counts inflating the binomial p-value (over-estimation). The list of genes that were manually curated, their change in XCI status, and the criteria for their manual classification can be found in *Figure 3—figure supplement 1* and *Supplementary file 1E*.

### Copy number analysis

Copy number analysis was performed using the R package, QDNASeq (*Scheinin et al., 2014*). Briefly, whole genome sequencing data for all three nmXCI females were mapped to the human reference genome (hg38) by BWA-MEM (*Li and Durbin, 2010*) using default parameters. Raw copy numbers were then estimated by counting the number of reads mapped to non-overlapping bins of 15 kb unless stated otherwise.

### Investigating chromosome 17p and 17q arm expression

Transcriptome mapping was performed using Salmon v1.9.0 (`--gcBias --seqBias --numBootstraps` 100) (*Patro et al., 2017*) using known Refseq transcripts (NM_* and NR_*) from the GRCh38. p12 assembly (GCF_000001405.38_GRCh38.p12_rna.fna.gz) as a reference. Library type options were IU. Raw FASTQ reads were quality trimmed using FastP (v.0.20.0) with default settings. Abundance estimates were converted to h5 format using Wasabi (https://github.com/COMBINE-lab/wasabi, *Patro et al., 2019*). Sleuth was used for data normalization and normalized gene abundances (*Pimentel et al., 2017*; *Yi et al., 2018*). Expression matrix (TPM) can be found in *Supplementary file 1H*.

### Acknowledgements

We thank Shadi Jafari and Sandra Hellberg for helpful discussions and input on the manuscript. Work in the lab of Colm Nestor was funded by the Swedish Research Council (2020–01277_VR) and the European Research Council (XX-Health-101045171).The project was funded by the Swedish Research Council (2020–01277_VR) and the European Research Council (XX-Health-101045171).

## Additional information

### Funding

| Funder | Grant reference number | Author |
|---|---|---|
| European Research Council | XX-Health-101045171 | Colm E Nestor |
| Vetenskapsrådet | 2020-01277_VR | Colm E Nestor |

The funders had no role in study design, data collection and interpretation, or the decision to submit the work for publication.

### Author contributions

Bjorn Gylemo, Data curation, Software, Formal analysis, Visualization, Methodology, Writing - original draft, Writing - review and editing; Maike Bensberg, Conceptualization, Visualization, Methodology, Writing - original draft, Writing - review and editing; Colm E Nestor, Conceptualization, Supervision, Funding acquisition, Investigation, Visualization, Writing - original draft, Project administration, Writing - review and editing

### Author ORCIDs

Bjorn Gylemo http://orcid.org/0000-0001-5253-6737
Maike Bensberg https://orcid.org/0000-0003-2395-6083
Colm E Nestor https://orcid.org/0000-0001-5853-1769

Reviewer #1 (Public review): https://doi.org/10.7554/eLife.102701.3.sa1
Reviewer #2 (Public review): https://doi.org/10.7554/eLife.102701.3.sa2
Reviewer #3 (Public review): https://doi.org/10.7554/eLife.102701.3.sa3
Reviewer #4 (Public review): https://doi.org/10.7554/eLife.102701.3.sa4
Author response https://doi.org/10.7554/eLife.102701.3.sa5

## Additional files

### Supplementary files

MDAR checklist

Supplementary file 1. Supplementary tables. (A) Genes used in screening to identify non-mosaic XCI (nmXCI) females. Compiled from Supplementary Table 13 in *Tukiainen et al., 2017*, removing genes with a variable escape status in UPIC (XCI across tissues ! = 'Partial, heterogeneous') and genes residing in the pseudoautosomal region (Region ! = 'PAR'). (B) Allelic expression of the genes selected in Supplementary File 1A for all 285 females analyzed from Genotype-Tissue Expression (GTEx). (C) Allelic expression of the genes selected in Supplementary File 1A for all tissues available for the nmXCI females: UPIC, nmXCI-1 (13PLJ), and nmXCI-2 (ZZPU). (D) Allelic expression data of 380 genes on the X-chromosome for each of the nmXCI females UPIC: nmXCI-1 (13PLJ) and nmXCI-2 (ZZPU). (E) List of genes which were manually curated, the directionality of the manual curation, the reasoning behind the manual curation. (F) Sample- and tissue-IDs for all data analyzed for nmXCI females: UPIC, nmXCI-1 (13PLJ), and nmXCI-2 (ZZPU). (G) Tissue abbreviations. (H) Gene expression (TPM) matrix of all tissues for nmXCI females: UPIC, nmXCI-1 (13PLJ), and nmXCI-2 (ZZPU).

### Data availability

All data and analysis scripts are included in this manuscript or are available on GitHub https://github.com/ColmNestorLab/tissue_XCI (copy archived at *ColmNestorLab, 2024*).

The following previously published dataset was used:

| Author(s) | Year | Dataset title | Dataset URL | Database and Identifier |
|---|---|---|---|---|
| GTEx Consortium | 2013 | Common Fund (CF) Genotype-Tissue Expression Project (GTEx) | https://www.ncbi.nlm.nih.gov/projects/gap/cgi-bin/study.cgi?study_id=phs000424.v1.p1 | dbGAP, phs000424.v1.p1 |

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
