## [Editor Report · eLife Assessment]

The study provides a **valuable** analysis of escape from X-inactivation based on three rare female GTEX-donors with non-mosaic X-inactivation. The methods and analyses are **solid** and broadly support the authors' claims. Their data are more comprehensive than those presented previously and add significant weight to evidence for which genes are inactivated or escape from X inactivation in humans.

---

## [Referee Report · Reviewer #1 (Public review)]

Summary:

This manuscript investigates genes that escape X-Chromosome Inactivation (XCI) across human tissues, using females that exhibit skewed or non-random XCI. The authors identified 2 female individuals with skewed XCI in the GTex database, in addition to the 1 female skewed sample in this database that has been described in a previous publication (Ref.16). The authors also determined the genes which escape XCI for 380 X-linked genes across 30 different tissues.

Strengths:

The novelty of this manuscript is that the authors have identified the XCI expression status for a total of 380 genes across 30 different human tissues, and also discovered the XCI status (escape, variable escape, or silenced) for 198 X-linked genes, whose status was previously not determined. This report is a good resource for the field of XCI, and would benefit from additional analyses and clarification of their comparisons of XCI status.

---

## [Referee Report · Reviewer #2 (Public review)]

Summary:

Gylemo et al. present a manuscript focused on identifying the X-inactivation or X-inactivation escape status for 380 genes across 30 normal human tissues. X-inactivation status of X-linked genes across tissues is important for understanding sex-specific differences in X-linked gene expression and therefore traits, and the likely effect of X-linked pathogenic variants in females. These new data are significant as they double the number of genes that have been classified in the human, and double the number of tissues studied previously.

Strengths:

The strengths of this work are that they analyse 3 individuals from the GTex dataset (2 newly identified, 1 previously identified and published) that have highly/ completely skewed X inactivation, which allows the study of escape from X inactivation in bulk RNA-sequencing. The number of individuals and breadth of tissues analysed adds significantly to both the number of genes that have been classified and the weight of evidence for their claims. The additional 198 genes that have been classified and the reclassification of genes that previously had only limited support for their status is useful for the field.

In analysing the data they find that tissue-specific escape from X inactivation appears relatively rare. Rather, if genes escape, even variably, it tends to occur across tissues. Similarly if a gene is inactivated, it is stable across tissues.

Comments on revised version:

The authors have answered all of my queries. While they have not been able to pinpoint the genetic cause of the highly skewed XCI cases in their cohort, I agree this is beyond the scope of this study. I have no further requests.

---

## [Referee Report · Reviewer #3 (Public review)]

Summary:

Nestor and colleagues identify genes escaping X chromosome inactivation (XCI) in rare individuals with non-mosaic XCI (nmXCI) whose tissue-specific RNA-seq datasets were obtained from the GTEX database. Because XCI is non-mosaic, read counts representing a second allele are tested for statistical significant escape, in this case > 2.5% of active X expression. Whereas a prior GTEX analysis found only one nmXCI female, this study finds two additional donors in GTEX, therefore expanding the number of assessed X-linked genes to 380. Although this is fewer than half of X-linked genes, the study demonstrates that although rare, nmXCI females are represented in RNA-seq databases such as GTEX. Therefore this analytical approach is worthwhile pursuing in other (larger) databases as well, to provide deeper insight into escape from XCI which is relevant to X-linked diseases and sex differences.

Strengths:

The analysis is well-documented, straight-forward and valuable. The supplementary tables are useful, and the claims in the main text well-supported.

Weaknesses:

There are very few, except that this escape catalogue is limited to 3 donors, based on a single (representative) tissue screen in 285 female donors, mostly using muscle samples. However, if only pituitary samples had been screened, nmXCI-1 would have been missed. Additional donors in the 285 representative samples cross a lower threshold of AE = 0.4. It would be worthwhile to query all tissues of the 285 donors to discover more nmXCI cases, as currently fewer than half of X-linked genes received a call using this very worthwhile approach.

Comments on revised version:

The authors incorporated some textual changes, but deferred any new analysis, or expansion from these two new skewed donors to include more individuals/tissues, or going more in depth for individual genes to future manuscripts. They appear to have that option at eLife.

---

## [Referee Report · Reviewer #4 (Public review)]

Summary:

This study by Gylemo et al. investigates genes that escape X-Chromosome Inactivation (XCI) by analyzing RNA-sequencing data from three female individuals with highly skewed XCI identified in the GTEx database-two newly reported and one previously described. Utilizing these rare non-mosaic XCI cases, the authors assess allelic expression patterns across 30 normal human tissues to classify the XCI status of 380 X-linked genes, including 198 not previously annotated. The study provides a broader and more comprehensive catalog of XCI escape, contributing valuable insights into sex-specific gene expression and the potential implications of X-linked variants in disease.

Strengths:

The primary strength of this work lies in its expanded scope: it doubles the number of tissues and significantly increases the number of X-linked genes with known XCI status compared to previous studies. By focusing on rare individuals with non-random XCI, the authors provide a unique opportunity to observe allelic expression and classify escape status with more confidence. Their findings that escape from XCI is relatively consistent across tissues (rather than tissue-specific) enhance the understanding of XCI mechanisms. The methodology is robust, the data are well-documented, and the supplementary resources are comprehensive. This study thus represents a valuable resource for the XCI field and a promising basis for future investigations.

Weaknesses:

Despite its strengths, the study is limited by its reliance on only three individuals, which restricts statistical power and generalizability. Concerns were raised regarding the comparability of XCI status across tissue types and cell lines, particularly given that previous classifications may have used cancer or immortalized cells. Additionally, more could be done to explore the genetic basis behind the observed skewed XCI, which might affect the conclusions about escape patterns. Finally, the authors are encouraged to expand their approach to additional RNA-seq datasets or single-cell analyses to validate their findings and potentially discover more individuals with skewed XCI, which would deepen understanding of this important biological phenomenon.

---

## [Author Response]

The following is the authors’ response to the original reviews

**Public Reviews:**

**Reviewer 1 (Public review):**
(1) The authors state that they have reclassified the allelic expression status of 32 genes (shown in Table S5, Supplementary Figure 3). The concern is the source of the tissue or cell line which was originally used to make the classification of XCI status, and whether the comparisons are equivalent. For example, if cell lines (and not tissues) were used to define the XCI status for EGFL6, TSPAN6, and CXorf38, then how can the authors be sure that the escape status in whole tissues would be the same? Also, along these lines, the authors should consider whether escape status in previous studies using immortalized/cancer cell lines (such as the meta-analyses done in Balaton publication) would be different compared to healthy tissues (seems like it should be). Therefore, making comparisons between healthy whole tissues and cancer cell lines doesn't make sense.

Indeed, many previous classifications were based on clonal cell lines, which could result in atypical patterns of escape due to the profound and varied effects of adaptation to culture. However, one of the primary goals of our study was to directly determine allele-specific expression from the X-chromosome in healthy primary tissues, in part to exclude the potential confounding effects of cell culture.

Whereas we do perform comparisons with cell culture-based classifications, we also provide detailed comparisons with the previous classification of Tukiainen *et al*, which also uses primary human tissues. In addition, whereas the comparison with Balaton *et al* is not optimal, we hold that it is valuable as it reveals which genes may exhibit aberrant escape patterns in culture. Finally, despite the above reservations, our comparison revealed an over-whelming agreement with previous research which suggests that in the vast majority of cases, escape appears to be correctly maintained in culture.

(2) The authors note that skewed XCI is prevalent in the human population, and cite some publications (references 8, 10-12). If RNAseq data is available from these female individuals with skewed XCI (such as ref 12), the authors should consider using their allelic expression pipeline to identify XCI status of more X-linked genes.

Indeed, we completely agree and are in the process of obtaining this data which has proven complex and time-consuming in the currently regulatory environment.

(3) It has been well established that the human inactive X has more XCI escape genes compared to the mouse inactive X. In light of the author's observations across human tissues, how does the XCI status compare with the same tissues in mice?

This is a very interesting point, and a comparison we are currently working on. However, this is a major undertaking and one that is outside of the scope of this study. We do appreciate the differences in mice and humans on X-chromosome level and could only speculate on the overlap being relatively small as the number of escapees in mice has been shown the be far lower than in humans.

**Reviewer 2 (Public review):**
In my view there are only minor weaknesses in this work, that tend to come about due to the requirement to study individuals with highly skewed X inactivation. I wonder whether the cause of the highly skewed X inactivation may somehow influence the likelihood of observing tissue-specific escape from X inactivation. In this light, it would be interesting to further understand the genetic cause for the highly skewed X inactivation in each of these three cases in the whole exome sequencing data. Future additional studies may validate these findings using single-cell approaches in unrelated individuals across tissues, where there is normal X inactivation.

We thank the reviewer for their positive assessment of our work. This is a point we have and continue to grapple with. We cannot rule out that the genetic cause of complete skewing may influence tissue-specific XCI. Moreover, the genetic cause for the non-mosaic XCI is currently unclear and is likely to vary between individuals, which could also result in inter-individual variation in tissue-specific escape. We are currently performing large prospective studies in the tissues of healthy females to specifically address this point.

**Reviewer 3 (Public review):**
There are very few, except that this escape catalogue is limited to 3 donors, based on a single(representative) tissue screen in 285 female donors, mostly using muscle samples. However, if only pituitary samples had been screened, nmXCI-1 would have been missed. Additional donors in the 285 representative samples cross a lower threshold of AE = 0.4. It would be worthwhile to query all tissues of the 285 donors to discover more nmXCI cases, as currently fewer than half of X-linked genes received a call using this very worthwhile approach.

We thank the reviewer for their positive assessment of our work. Of course, we agree that a tissue-wide screen in all individuals would have been optimal and is a line of research we are currently pursuing. However, the analysis of allele-specific expression in all 5,000 RNA-seq samples is a massive undertaking and was simply not practicable within the time-scale of this study.

**Recommendations for the authors:**

**Reviewer #2 (Recommendations for the authors):**
Thanks to the authors for an interesting manuscript! I enjoyed reading it and the care that has gone into explaining the analyses and the findings. There are a few recommendations that I have for strengthening the work.

We thank the reviewer for the nice feedback. Much appreciated.

(1) I would like to see a genetic analysis of the three individuals, to try and identify the genetic causes of the skewed X inactivation beyond just considering the XIC or translocations. The cause of the highly skewed X inactivation would be of interest to many.

This is certainly a very interesting avenue of research and one that we are currently focusing on. However, in the current study we simply had too few skewed XCI females to assess this in an exhaustive manner. To tackle this issue, we have begun a prospective study of healthy females to identify additional non-mosaic females.

(2) I wonder whether the cause of the skewed XCI may somehow influence the assessment of tissue-specific escape? If there is a problem with X inactivation itself, perhaps escape would also be different, making it appear more constitutive than tissue-specific?

This is a point we have and continue to grapple with. We cannot rule out that the genetic cause of complete skewing may influence tissue-specific XCI. Moreover, the genetic cause for the non-mosaic XCI is currently unclear and is likely to vary between individuals, which could result in inter-individual variation in tissue-specific escape.

(3) Presentation/wording suggestions:I think the abstract is likely a bit inaccessible to those outside the field. I am in the X inactivation field, but don't use the term non-mosaic X inactivation, but rather would call it highly skewed, or non-random X inactivation. In my view, it would be simpler for the abstract to call non-mosaic XCI highly skewed XCI instead, or to use more words to ensure it is clear for the reader.

We agree that the terminology of completely skewed/non-mosaic XCI could be more clearly defined in the abstract and have clarified this. “Using females that are non-mosaic (completely skewed) for X-inactivation (nmXCI) has proven a powerful and natural genetic system for profiling X-inactivation in humans.”

I would consider calling the always escape genes constitutive escapees, while the variable may be facultative.

This is something we have also considered and have received differing feedback on. However, we will definitely keep this in mind for future publications.

Line 132, it would be useful to explain median >0.475 as less than 2.5% of reads coming from the inactive allele here, not just in the methods. Can you also explain why this cutoff was chosen?

We thank the reviewer for this clarification. A clarification has been added to the main text as suggested.

The cutoff was applied to account for potential variations in skewing, given that we screened only a single tissue sample per individual. Although nmXCI females are theoretically expected to have 0% of reads originating from the 'inactive' allele, this is not always observed due to (a) technical errors such as PCR or sequencing inaccuracies, or (b) differences in skewing between tissue types.

Lines 156-160 describe how the heterozygous SNPs were identified in relation to Figure 2. I read these in the methods so that I could understand Figure 1, so I suggest moving this section up.

We have moved the section as suggested by the reviewer.

Line 156, consider adding in a sentence to describe what is shown in Figures 2A and B i.e, the overlap of SNPs and spread along the X.

We have added a sentence describing what is shown in Figures 2A and 2B as suggested by the reviewer.

Line 217, it would be useful to give the % of genes that show tissue-specific escape, to quantify rare.

We have added a sentence quantifying ‘rare’ at the suggested line.

(4) Typos:Line 119, missing 'the most' before extensive (and remove an).

We thank the reviewer for pointing this out. This error has been corrected.

**Reviewer #3 (Recommendations for the authors):**
Some results in the supplementary figures were quite striking. What is going on with DDX3X and ZRSR2? How come total read counts are so different between individuals?

Indeed, this is a very intriguing observation and one that we have simply failed to understand thus far. We are currently performing a large prospective study to obtain greater number of non-mosaic females and tissues samples. Hopefully, additional observations across females will allow us to gain further insights into the inter-individual behaviour of DDX3X and ZRSR2.

One item I would like to see added is some analysis to address the cause of these extremely skewed XCI individuals. The copy number analysis suggests there are some segmental deletions on the X in all three nmXCI cases. Where are these deletions, and do any fall in the region of the X-inactivation centre? Have the authors performed any analysis of potentially deleterious X-linked variants in the WGS or WES data? Why are these donors so skewed? It's interesting that UPIC was still more skewed than the other two.

The segmental deletions the reviewer points out are not segmental deletions, the same variation in coverage is found in all females we’ve looked at including females with a mosaic XCI (see Author response image 1 below where the same pattern of slightly lower read counts is observed at the same sites in all female samples). No deletions were identified in the XIC region. No analysis was performed of deleterious X-linked variants. Why the donors are so skewed is unknown and intriguing. Indeed, identifying the origin of extreme skewing (including the females in this study) is now the main focus of the group. Whereas UPIC had trisomy 17, which has likely resulted in the observed skewing, we have not yet found a genetic variant that could explain the skewing observed in 13PLJ or ZZPU.

**Author response image 1. sa5fig1:** Copy number as log2 ratio using 500kb bins across the X-chromosome for 3 mosaic XCI females (1QPFJ, OXRO, and RU1J) and 3 nmXCI females, UPIC, nmXCI-1 and nmXCI-2.

This is not necessary to address with new analyses, but as alluded to above, the authors could screen more than a single representative tissue. And to apply this analysis to larger databases (UK biobank), which the authors may be planning to do already.

This an avenue of research we are currently investigating.

The code is well-documented and accessible. Additional information on the manual reclassification (to deal with inflated binomial P-values) would be helpful. Why not require a minimal threshold for escape (10% of active X allele) in addition to a significant binomial P (inactive X exp. > 2.5% of active)?

We thank the reviewer for this positive assessment of the code.

Indeed, how to define ‘escape’ is a vexed issue, and one we feel has been given undue weight within the field. In reality, studies of escape are often dealing with sparse data (e.g. read depth), few observations (genes and individuals) and substantial amounts of missing data. Thus, it is unlikely that a standard statistical approach will be sensitive and specific across different studies and data types. Similarly, cut-offs, though useful would also need to be adjusted to the data type and quality in any given study.

Whereas we initially used a significant binomial P-value as our sole test (often quoted as ‘best practice’), this resulted in wide-spread inflation of P-values. Thus, we switched to manually curating the allelic expression status of all 380 genes using the empirical guideline of allelic ratio >0.4 (also a commonly used cut-off) as indicating mono-allelic expression. We considered combining the binomial P-value with the cut-off but felt that this would result in an overly complex definition of escape and would unnecessarily exclude many genes from classification, due to the opposing effects of low/high read depth on the binomial and cut-off approaches respectively.

Indeed, due to the difficultly of both accurate and objective ‘classification’ of escape that we placed an emphasis on clearly displaying all data for each gene in each individual to allow readers to see all the data on which each classification was based.